# Unity in Diversity and Diversity in Unity—Vaccination Policies in EU Countries

**DOI:** 10.3390/healthcare13010019

**Published:** 2024-12-25

**Authors:** Elisaveta Petrova-Geretto, Alexandrina Vodenitcharova, Guenka Petrova

**Affiliations:** 1Department of Bioethics, Faculty of Public Health, Medical University-Sofia, 1431 Sofia, Bulgaria; e.geretto@foz.mu-sofia.bg (E.P.-G.); a.vodenicharova@foz.mu-sofia.bg (A.V.); 2Department of Organization and Economy of Pharmacy, Faculty of Pharmacy, Medical University-Sofia, 1504 Sofia, Bulgaria

**Keywords:** vaccination, policy, European Union, European countries, ethics

## Abstract

Background/Objectives: This article emphasizes the comprehensive importance of vaccination, exploring its role in disease prevention, addressing growing concerns around vaccine hesitancy, and underscoring the crucial need for high vaccination coverage rates. Methods: Our review examines EU-level and national policies on vaccination, utilizing EU sources, with a specific focus on regulatory and policy documents. Vaccination calendars in the EU were reviewed through the ECDC Vaccine Scheduler webpage. The parameters of this literature review were further selected in collaboration with an Emory Health sciences librarian using the following search terms: healthcare providers, communication, public health, informed consent, and bioethics. Results: The analysis of limited articles on community engagement, moral and political philosophy, and public health ethics informs the ethical consideration of informed consent in public health interventions. Countries exhibit varying relationships between their vaccination programs and society, with technocratic and populist influences shaping vaccination policies. Conclusions: Europe demonstrates diversity in vaccination policies, with availability, funding, and scheduling reflecting distinct approaches to vaccination.

## 1. Introduction

The 20th and 21st centuries have witnessed remarkable advancements in medical science, and vaccination stands as one of the most illustrious achievements in this journey towards public health excellence [1,2]. The effectiveness of vaccination in preventing communicable diseases cannot be overstated, and it stands as one of the most powerful and cost-effective public health interventions globally [3]. Annually, vaccination averts between 3.5 and 5 million deaths from vaccine-preventable infectious diseases [4]. Moreover, vaccination has achieved milestones like the eradication of smallpox and is currently on track to eliminate poliomyelitis [5,6]. It has also led to the elimination of diseases like measles, influenza type B, yellow fever, cholera, and hepatitis in various regions [7]. However, the lifesaving potential of vaccination can only be harnessed through the collective responsibility of achieving herd immunity, which is reliant on maintaining high vaccination coverage rates [1,8]. Herd immunity plays a pivotal role in controlling epidemics of communicable diseases as it not only safeguards those unable to receive vaccinations due to medical conditions but also reduces the risk of infection among susceptible individuals. From a societal point of view, herd immunity embodies our collective responsibility to one another [9] and represents an act of solidarity to our fellow human beings. The popular battle cry during the 2019–2022 pandemic “No one is safe until everybody is safe” epitomizes the core of the herd immunity concept [10,11,12]. It is worth pointing out that for some infectious diseases, especially for tetanus, and to a lesser degree for hepatitis B and SARS-CoV-2, this concept is not the leading argument. In these cases, self-protection is the main goal, but society still has a responsibility to raise awareness.

The responsibility to protect citizens is fundamental to the nation-state, which is intrinsically related to national memory, political history, culture, and political philosophy; thus, national governments must ensure the availability and affordability of vaccines, irrespective of their vaccination policies [10]. Different nations exhibit varying relationships between their vaccination programs and society, with various non-medical and non-scientific influences, which might suggest that politics, more so than science, shapes national vaccination policies [13,14]. The balance between the shared competence of the European Union (EU) and the autonomy of individual member states in healthcare is exemplified by the diversity in vaccination policies across the EU. Vaccinations are recommended or mandated to various degrees, with some countries recently adopting mandatory vaccination policies to improve coverage rates. Some countries have introduced mandatory vaccination measures to address vaccine-related concerns [15]. Additionally, the availability and funding of vaccines, as well as the number and schedule of doses, differ across countries, reflecting distinct approaches to vaccination. Currently, several European countries are grappling with outbreaks of vaccine-preventable diseases due to insufficient vaccination coverage rates caused by complex, overlapping factors, with disinformation and vaccine hesitancy being the most pressing issues. Vaccine hesitancy, as defined by the WHO, refers to “delay in acceptance or refusal of vaccines despite availability of vaccination services” [16,17,18].

The goal of the study is to provide an overview of the strategies employed by EU countries and institutions to promote vaccination, as well as the various regulatory frameworks governing vaccination across EU member states. Additionally, this study aims to compare how different EU countries approach vaccination policies concerning healthcare professionals, i.e., whether vaccination is mandatory, voluntary, or recommended for them.

## 2. Materials and Methods

A policy review was undertaken with a focus on EU-level and national policies, using the webpages of the European Commission, the European Centre for Disease Prevention and Control (ECDC), and European-wide professional organizations of physicians. Regulatory and policy documents were identified and analyzed. The review was guided by three principal research questions: whether a unified approach to vaccination exists among EU member states; what vaccinations were mandatory or recommended; and whether there were specific provisions for healthcare professionals. As the primary focus, EU websites and relevant policy documents were identified; a systematic literature review approach was not applied.

However, further search parameters were refined with the collaboration of a librarian from Emory Health Sciences. The following specific search terms were identified and utilized: “healthcare providers”, “communication”, “public health”, “informed consent”, and “bioethics”. The results of this search yielded a limited number of articles: “community engagement” returned 3 articles, “moral and political philosophy” yielded 9 articles, “informed consent in public health interventions” produced 7 articles, “public health ethics” resulted in 4 articles, and “informed consent in public health” yielded 6 articles.

Vaccination calendars in the EU were explored through a review of the ECDC Vaccine Scheduler webpage [13] and revisited for updates in September 2024 and November 2024. The available vaccination policies were examined and analyzed to provide an overview of national vaccination approaches in the EU and to discuss informed consent in public health from an ethical standpoint.

Throughout the article, “mandatory vaccination” is used to refer to a requirement and/or policy enforced by law [8], and it implies direct or indirect consequences for non-compliance [19].

## 3. Results

### 3.1. Policy Measures at National Level

Table 1 summarizes the vaccination policies in EU member states (27 countries + Liechtenstein) based on the information provided on the ECDC Vaccine Scheduler webpage [20]. The policies are a combination of mandatory policies, which “compel vaccination by direct or indirect threats of imposing restrictions in cases of non-compliance” [9,19] (p. 1), and recommendations and specific requirements for healthcare professionals.

The results indicate that half of the countries have mandatory vaccination policies and the other half have none, with the exception of Austria, which introduced non-mandatory vaccination recently. Countries differ in the details of their policies. Some countries rely solely on recommendations to encourage vaccination, while others enforce mandatory vaccination policies. The scope of mandatory vaccination policies and the consequences for parents who do not adhere to them also differ significantly among nations. In certain countries (France, Germany), parents are required to provide proof of their child’s vaccination against specific diseases as a prerequisite for enrollment in educational facilities (kindergarten and/or school). In contrast, other nations (Italy, Germany for measles) impose fines when parents refuse vaccinations for their children as a mechanism to encourage vaccination compliance. However, fines and penalties are rarely enforced as “in some cases penalties are only theoretical and never applied” [8] (p. 2). Previous studies [22,23] have found an association between mandatory vaccination, the magnitude of fines, and higher vaccination coverage. While most western European countries primarily offer recommended vaccinations, there is a notable prevalence of mandatory vaccination policies in Eastern Europe. Nevertheless, some Western European countries, including France and Italy (since 2017), as well as Germany, have undergone changes to their vaccination programs, incorporating one or more mandatory vaccinations to bolster vaccination coverage rates. The adoption of mandatory vaccination policies is influenced by various factors, including the prevalence of vaccine-preventable diseases, public health concerns, and political decisions. The decision to implement such policies is often grounded in the goal of achieving and maintaining high vaccination coverage rates to safeguard public health and curb the spread of infectious diseases. Greece maintains a policy of voluntary vaccination without any mandatory vaccinations in place for the general population.

The above overview exemplifies the complexity of vaccination policies across the EU, which include mandatory and recommended vaccination or a combination of both approaches with regional differences for federal states. Mandatory vaccination primarily targets children and prioritizes population-wide protection through the rigorous enforcement of mandatory vaccinations and comprehensive immunization programs. Other countries, however, rely on recommendations rather than mandates, respecting individual autonomy and personal responsibility in healthcare decisions.

With regard to healthcare professionals, similar conclusions were reached by the CPME, who undertook a survey of its members in 2018 regarding national policies with respect to specific professional groups, as well as the attitudes towards voluntary vs. recommended vs. mandatory vaccination [24]. The survey indicated wide-ranging variability among its members, with eleven medical associations returning feedback. For example, Malta emphasizes the importance of patient–doctor consent, relying on persuasion and easy access to vaccines. Switzerland focuses on ensuring vaccine availability for physicians and relies on successful voluntary schemes. In Romania, each health facility organizes annual influenza vaccination campaigns, and HCWs receive the influenza vaccine at their workplace, covered by the Ministry of Health budget. In Czechia, previous attempts at mandatory vaccination faced challenges due to transparency issues, leading to skepticism. In Germany, the medical association supports measures to boost vaccination coverage but opposes mandatory vaccination, emphasizing communication and raising awareness. The United Kingdom encourages vaccination through funding for practices, workplace vaccination, and incentives like additional leave days. Hungary underscores the importance of providing free vaccines for health professionals but notes that mandatory measures are ineffective. Finland provides for mandatory vaccination for HCPs; however, legislation allows non-immunized personnel to work only in exceptional cases. Denmark, Finland, and Sweden (the Nordic countries) provide national recommendations or guidelines for voluntary vaccination [21], while Austria supports mandatory vaccinations for healthcare professionals as a duty to avoid harming patients. Ireland offers free flu vaccination to healthcare professionals and encourages voluntary uptake. Each country’s approach reflects a balance between encouraging voluntary vaccination, respecting individual autonomy, ensuring vaccine availability, and considering the potential for mandatory measures. The effectiveness of these strategies varies, emphasizing the importance of tailored approaches to increase coverage among healthcare professionals. Thus, CPME members endorse measures aimed at boosting physicians’ vaccination rates but do not endorse mandatory vaccination as the ideal solution [24].

Although healthcare professionals are at high risk of contracting infectious diseases, most of the EU countries treat them as the general population. Others provide national recommendations and/or guidelines, such as the Nordic countries. France, Belgium, Slovenia (measles), and Italy (only for COVID-19) pose specific regulations for healthcare professionals.

### 3.2. Vaccination Strengthening Initiatives at European Level

Against this backdrop of diverse approaches to vaccination in the various EU member-states, the EU has undertaken a multifaceted approach to boost vaccination confidence and enhance immunization efforts across the continent. The strategy encompasses various initiatives and collaborations aimed at promoting vaccination as a critical public health tool.

One of the central elements of this approach is enhanced cooperation. The ECDC and the European Medicines Agency (EMA) have established a joint vaccine monitoring platform [25]. This platform facilitates the monitoring of vaccine safety and effectiveness, providing accurate and up-to-date information to ensure public trust in vaccines. The main purpose is to create a body of real-world evidence for vaccine safety and thus to minimize vaccine hesitancy.

Additionally, joint procurement of vaccines at the EU level ensures equitable access to vaccines for member states. Launched in 2014, The European Joint Procurement Agreement (JPA) [26] serves as a novel tool for coordinating the acquisition of vaccines and medicines in anticipation of pandemics. The primary goal of the JPA is to ensure fair and economically viable access to medical resources for all EU Member States involved, particularly in times of significant health emergencies.

Another cooperation platform is the Coalition for Vaccination, with leadership of the Coalition being shared by the Standing Committee of European Doctors (CPME), the European Federation of Nurses Associations (EFN), and the Pharmaceutical Group of the European Union (PGEU) [27]. The Coalition unites various stakeholders, including European healthcare professional associations, relevant student associations, and associated organizations focusing on public health and immunization. Established in 2019 by the European Commission, its main objectives include providing precise public information, dispelling misinformation, and facilitating the exchange of effective approaches. As summarised by Prof. Montgomery, immediate past President of the Standing Committee of European Doctors (CPME) and co-Chair of the Coalition, the three reasons behind vaccine hesitancy are “cautiousness, complacency, convenience” [17], and the four recommendations to overcome vaccine hesitancy are as follows: (1) transparency and information; (2) information campaigns to reach “the right people”; (3) bringing the vaccine to healthcare workers; (4) the introduction of occupational restrictions to be considered for unvaccinated healthcare workers [18]. Based on these recommendations, the Coalition published a “to-the-point” manifesto [19] during the COVID-19 pandemic to motivate, support, and encourage healthcare professionals to get vaccinated.

The EU-JAV (European Joint Action on Vaccination) is another initiative that brings together EU countries to collaborate on increasing vaccination uptake and building public confidence [27]. This joint effort focuses on a range of strategies, including behavioral insights and the monitoring of attitudes towards vaccination. During the Health Security Committee in Luxembourg in July 2021, an overview of the EU-JAV work and conclusions were presented: “the main determinant of suboptimal vaccine uptake is the lack of confidence in vaccine safety; regarding specific groups linked to suboptimal vaccine uptake, healthcare workers are listed as second group after immigrants; when it comes to target specific population groups, healthcare workers seems to be a low priority group; barriers to working on vaccine hesitancy and uptake are often related to organizational limits and lack of funding. EU-JAV main recommendations call for a need for better understanding on how to define vaccine hesitancy, a holistic approach rather than focus on vaccine confidence, better targeting of different population groups, and the need for long-term plans”.

Proactive communication plays a pivotal role in the EU’s vaccination confidence strategy. EU President Ursula von der Leyen calls for “Prioritis[ing] communication on vaccination, explaining the benefits and combating the myths, misconceptions and scepticism that surround the issue” for the “United in Protection” campaign, along with annual contributions to the European Immunization Week, which aims to raise awareness about the importance of vaccines [28]. To provide transparent and reliable information on vaccines, the EU has developed resources such as the European Vaccination Information Portal (vaccination-info.eu) and the Safe COVID-19 vaccines for Europeans webpage [29,30]. These platforms offer objective, evidence-based information to address concerns and questions regarding vaccines. Additionally, addressing misinformation and disinformation is a priority and funding under EU4Health and Horizon 2020 focuses on research countering vaccine misinformation and supporting disadvantaged and socially excluded groups. These initiatives reflect the EU’s commitment to fostering vaccination confidence, protecting public health, and achieving equitable vaccine distribution [31].

## 4. Discussion

In this study, we systematized vaccination policies in EU and EEA countries, as well as emerging and well-established policy approaches at the European level to improve vaccination coverage. We also reviewed measures applicable to healthcare professionals. It has been recognised that the attitudes, personal choices, awareness, knowledge about vaccines, and the vaccination of physicians, as well as a national policy approach towards the vaccination requirements of healthcare professionals, are some of the key factors for increasing vaccine confidence and thus vaccine uptake in the general population. In fact, the State of Vaccine Confidence in the EU + UK 2020 report concluded that “A strong association is found between the percentage of the public with high confidence in the safety and effectiveness of vaccines and the percentage of healthcare professionals with high confidence” [32]. The Maltezou, H.C. et al. study on “Vaccination policies for health-care workers in acute health-care facilities in Europe to inform on the existing policies on recommended and mandatory occupational vaccinations for healthcare professionals in Europe” provides context for this phenomenon. The study identifies “significant disparities and variations” [33] and concludes that this spectrum of diversity should stimulate a debate for re-evaluating vaccination policies from the standpoint of occupational health risk management in order to improve healthcare professional and patient safety. Karafilakis et al. recognize that the attitudes of healthcare workers (HCWs) towards vaccination play a crucial role in the successful implementation of vaccination programs as recommendations from healthcare providers strongly influence vaccination decisions [34]. Their qualitative study revealed vaccine hesitancy among HCWs in countries like Croatia, Romania, France, and Greece. In France, for example, it was found that a notable percentage of general practitioners (GPs) only sometimes or never recommend specific vaccines to their patients. The majority of GPs surveyed in Czechia and Bulgaria would not recommend the seasonal influenza vaccine to pregnant women, and only 68% of GPs surveyed in Czechia, 74% in Slovakia, and 75% in Bulgaria would recommend the HPV vaccine, which is the lowest among all 28 countries [34].

With knowledge, science, medicines, procedures, political will, and international cooperation available, it becomes an issue of justice and fairness to collectively tackle public health challenges. Upholding individual autonomy and freedom of choice in public health interventions in fact diminishes, if not entirely limits, individual freedom and choice on a personal level because of the consequences of ill-health [35,36].

The anti-reductionist point highlights that as a part of public health, the health status of the people is more than the simple aggregation of individual conditions, and that a truly public emphasis and lens should be kept on public health interventions and should be the focus when discussing vaccination and vaccination policies [37].

From a bioethical perspective, the issue on social responsibility and health is discussed in Art. 14 from the Universal Declaration of Bioethics and Human. It calls for healthcare systems to ensure equitable distribution and access to vaccines. Further, it encourages societies to prioritize the common good, emphasizing that collective efforts in vaccination contribute to healthier and more resilient communities [38,39,40]. The ethical foundation of healthcare professionals rests on their commitment to disseminate accurate and comprehensible information regarding the benefits and risks of vaccinations. The duty to provide evidence-based scientific advice emphasizes the importance of healthcare professionals staying abreast of current research and advancements in vaccinology. This ethical responsibility reflects a commitment to the principles of autonomy and beneficence, recognizing the right of individuals to make decisions about their health while ensuring that these decisions are well-informed and conducive to the greater good. In essence, the bioethical integration of social responsibility and health necessitates a continuous dedication to providing evidence-based scientific advice, fostering a culture of trust, and promoting public health through informed decision-making in vaccination. As some authors have pointed out, there are global ethical considerations that should be taken into account when vaccinating children, especially with novel vaccines [41]. Such a recommendation needs precise, scientifically grounded information on the associate risks, disease severity, and vaccine safety and effectiveness to evaluate the available alternatives, as well as the levels of coercion associated with each. From an ethical point of view, the decision should be carefully taken after considering the threat of the disease to human health, the comparative effectiveness of the vaccines, and the risk of the negative perception of coercion.

Vaccine hesitancy is an important ethical issue too because recent studies show that several EU countries are going through unprecedented outbreaks of vaccine-preventable diseases, and factors such as cultural and organizational background are important in overcoming this phenomenon [42,43]. Regulatory decisions, including the introduction of mandatory vaccination, should be justified by ensuring the appropriateness of such solutions in light of the existing problem, and that they do not risk unintended consequences. Measures at the European level are needed to overcome important barriers, such as the cost of vaccination, distance, and time, to achieve high levels of uptake, especially for marginalized or vulnerable populations. European institutions could also provide information in the form of public information campaigns or interactions between health workers and the population [44].

The limitation of this study is that the COVID-19 pandemic has not been discussed because the political debate over-influenced scientific and evidence-based policymaking and COVID-19 vaccine mandates. Additionally, we recognize the risk that some of the information at the ECDC might not reflect the current situation in the observed countries, but we do believe that at the time of the search, the information in the ECDC database was correct. Further analyses involving the national authorities need to be conducted to validate, in more detail, the ECDC information.

## 5. Conclusions

The comparative analysis of vaccination policies across European countries underscores the diversity of approaches within the broader context of public health and individual choice. Each nation’s policy reflects its unique cultural, legal, and healthcare considerations while striving to protect public health through vaccination. The nuanced strategies employed highlight the importance of tailored approaches that align with the specific needs and values of each society.

## Figures and Tables

**Table 1 healthcare-13-00019-t001:** EU countries’ vaccination policies.

Country	Mandatory	Vaccination Requirements for Health Professionals
Austria	As of March 2023, there are no mandatory vaccinations in Austria	Not mandatory per se but there is a moral expectation to do so; internally addressed by medical establishments
Belgium	Yes Children—Poliomyelitis	Specific vaccination regulations for medical personnel
Bulgaria	Yes Children—Tubercolosis, Diphtheria, Tetanus, Pertussis, Poliomyelitis, Haemophilus Influenzae type B, Hepatitis B, Pneumococcal disease, Measles, Mumps, and Rubella Adults—Diphtheria and Tetanus	As general population (Bulgarian immunization calendar)
Croatia	Yes Children—Tubercolosis, Diphtheria, Tetanus, Pertussis, Poliomyelitis, Haemophilus Influenzae type B, Hepatitis B, Pneumococcal disease, Measles, Mumps, and Rubella Adults—Tetatus	As general population (mass immunization calendar)
Cyprus	None	Recommended
Czechia	Yes Children—Tubercolosis, Diphtheria, Tetanus, Pertussis, Poliomyelitis, Haemophilus Influenzae type B, Hepatitis B, Measles, Mumps, and Rubella Adults—Measles (catch-up), Hepatitis B, Hepatitis A (above 60), and Tetatus	As general population
Denmark	None	National recommendations/guidelines for healthcare worker immunization [21]
Estonia	None	As general population
Finland	None	National recommendations/guidelines for healthcare worker immunization [21]
France	Yes Children—Diphtheria, Tetanus, Pertussis, Poliomyelitis, Hemophilus Influenzae type B, Hepatitis B, Pneumococcal disease, Measles, Mumps, and Rubella	Yes, healthcare staff are subject to mandatory vaccinations against specific diseases, with exemptions allowed for medical reasons
Germany	Yes Children—Measles	Yes (measles)
Greece	None	As general population
Hungary	Yes Children—Tubercolosis, Diphtheria, Tetanus, Pertussis, Poliomyelitis, Haemophilus Influenzae type B, Hepatitis B, Pneumococcal Disease, Measles, Mumps, Rubella, and Varicella	Yes
Ireland	None	Strongly recommended not mandatory
Italy	Yes Children—Diphtheria, Tetanus, Pertussis, Poliomyelitis, Haemophilus Influenzae type B, Hepatitis B, Measles, Mumps, Rubella, and Varicella	Yes, against COVID-19
Latvia	Yes Children—Tubercolosis, Rotavirus infection, Diphtheria, Tetanus, Pertussis, Poliomyelitis, Haemophilus Influenzae type B, Hepatitis B, Pneumococcal Disease, Measles, Mumps, Rubella, Varicella, and HPV Adults—Diphtheria and Tetanus	As general population
Liechtenstein	None	None
Lithuania	None	None
Luxemburg	None	None
Malta	Yes Children—Diphtheria, Tetanus, and Poliomyelitis	Recommended
Netherlands	None	None
Norway	None	National recommendations/guidelines for healthcare worker immunization [21]
Poland	Yes Children—Tubercolosis, Diphtheria, Tetanus, Pertussis, Poliomyelitis, Haemophilus Influenzae type B, Hepatitis B, Pneumococcal Disease, Measles, Mumps, and Rubella Adults—Diphtheria and Tetanus	Yes, COVID-19
Portugal	None	None
Romania	None	Recommended
Slovakia	Yes Children—Tubercolosis, Diphtheria, Tetanus, Pertussis, Poliomyelitis, Haemophilus Influenzae type B, Hepatitis B, Pneumococcal Disease, Measles, Mumps, and Rubella Adults—Diphtheria and Tetanus	None
Slovenia	Yes Yes Children—Tubercolosis, Diphtheria, Tetanus, Pertussis, Poliomyelitis, Haemophilus Influenzae type B, Hepatitis B, Measles, Mumps, and Rubella	Yes, the obligatory vaccinations against measles are specifically stated; healthcare workers have to receive two doses of the measles vaccine if they were not previously vaccinated before starting work or show natural immunity
Spain	None	None
Sweden	None	None

## Data Availability

There are no new data created in this study.

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
