# Peer review of "Unity in Diversity and Diversity in Unity—Vaccination Policies in EU Countries"

_healthcare, 2024, doi:10.3390/healthcare13010019_

Round 1
Reviewer 1 Report
Comments and Suggestions for Authors
Thank you for the chance to review this manuscript. Such comparative review is necessary for advocacy around vaccination policies and timely considering the drop in vaccine confidence post pandemic.
Whereas I think this manuscript is interesting and has merit, there are some concerns: both in style of writing but also in content (actual scoping review methodology and presentation of findings) that I feel the authors need to address before this can be ready for publication.
1.
2. Materials and methods: I am not certain whether a scoping review stepwise approach was employed. For example, how did you choose your team and what expertise did you have? What were your exclusion and inclusion criteria? Had a scoping review on this already been conducted? Consider a stepwise approach as outlined here: Steps for Conducting a Scoping Review - PMC (nih.gov)
3. Results:
a. Paragraph 1: please mention how many countries were included.
b. Table 1: please include what mandatory means or how you have defined mandatory for the purpose of this study, for what ages and what the mandate implies (e.g: to attend school etc)
c. Table 1 should really represent the articles and specifics of the articles identified. What are the articles used for review that the search has provided and what are the specifics of each article?
d. It would put this into perspective to also have the reported vaccination uptake rate of a few sentinel vaccines (such as diphtheria-tetanus-pertussis containing vaccine and measles containing vaccines) for each country and maybe health care worker influenza vaccine uptake for each country in Table 1.
e. Line 100: fines and penalties are rarely enforced: how do you know that, what is your reference for this?
f. 3.2. Policy measures at European level: I am uncertain how this section relates to the scoping review findings. Could the authors please outline how the findings such as “enhanced cooperation” came out of the scoping review? This section sounds to me a bit more suited to an editorial rather than actual findings of a scoping review. How did you come to these conclusions from the studies you looked at?
4. Discussion:
a. Lines 262-297: should these be part of the results rather than discussion?
b. “Social justice” was not hinted at or mentioned in the results: it is first mentioned in discussion: what are the authors drawing in as part of their results to make the link to the discussion?
c. I really cannot understand the connection between the results and the discussion and I think this whole section needs to be redone so that it matches the results. For example, the CPME survey findings: were they used as a part of the scoping review OR are they just used as part of the discussion and those findings not actually part of the results? If so, then why did the search not yield reference 26 (the CPME survey) as part of the scoping review sources?
Conclusion: What does this line mean: " There is no different approaches towards the vaccination policy for healthcare workers." Does it mean "this is not different than the approach for vaccination policy for health care workers?
Comments on the Quality of English Language
Please ensure you use the same tenses throughout and same passive or active voice throughout. For example Materials and methods has multiple tenses used, causing some confusion. Third person tenses are often incorrect and I wonder whether some past tenses were just missed in editing. There are also multiple grammatical errors throughout the document: please edit for grammar and tense use. Some simple examples that need correcting: you use both term immunisation as well as immunization: choose one. The whole manuscript needs a very detailed edit for appropriate language.
Author Response
Reviewer 1
Open Review
- Whereas I think this manuscript is interesting and has merit, there are some concerns: both in style of writing but also in content (actual scoping review methodology and presentation of findings) that I feel the authors need to address before this can be ready for publication.
Thank you for your overall positive opinion
- Materials and methods: I am not certain whether a scoping review stepwise approach was employed. For example, how did you choose your team and what expertise did you have? What were your exclusion and inclusion criteria? Had a scoping review on this already been conducted? Consider a stepwise approach as outlined here: Steps for Conducting a Scoping Review - PMC (nih.gov)
To better explain our approach, we made the following changes in the beginning of the methodology section:
Policy review was undertaken with a focus on EU-level and national policies, using the webpages of the European Commission, the European Centre for Disease Prevention and Control (ECDC), and European-wide professional organizations of physicians. Regulatory and policy documents were identified and analyzed. The review was guided by four principal research questions: whether a unified approach to vaccination existed among EU member states, what vaccinations were mandatory or recommended, and whether there were specific provisions for healthcare professionals. As the primary focus were policy documents and EU-websites and relevant policy documents were identified, systematic literature review approach was not applied.
However, further search parameters were refined with the collaboration of a librarian from Emory Health Sciences. The following specific search terms were identified and utilized: “health care providers”, “communication”, “public health”, “informed consent”, and “bioethics”. The results of this search yielded a limited number of articles: "community engagement" returned 3 articles, "moral and political philosophy" yielded 9 articles, "informed consent in public health interventions" produced 7 articles, "public health ethics" resulted in 4 articles, and "informed consent in public health" yielded 6 articles.
- Results:
- Paragraph 1: please mention how many countries were included.
We add the following clarification:
EU member states (27 countries+Liechtenstein) based on the information provided in ECDC Vaccine Scheduler webpage
- Table 1: please include what mandatory means or how you have defined mandatory for the purpose of this study, for what ages and what the mandate implies (e.g: to attend school etc)
We add the following clarification on line 97 from the results section: The policies are a combination of mandatory which ”compel vaccination by direct or indirect threats of imposing restrictions in cases of non-compliance” [21, 22 (p.1)] , recommended and specific requirements for healthcare professionals.
- Table 1 should really represent the articles and specifics of the articles identified. What are the articles used for review that the search has provided and what are the specifics of each article?
As stated, this is a policy review focusing on the differences in policy approach towards vaccination. Therefore Table 1 represents the major differences in vaccination policy across observed countries. We did several improvements of the table 1 to represent more clearly the differences in policies regarding children and aduts, as well as healthcare workers. For the policy analysis the major sources are described in the methodology section as: webpages of the European Commission, the European Centre for Disease Prevention and Control (ECDC)
- It would put this into perspective to also have the reported vaccination uptake rate of a few sentinel vaccines (such as diphtheria-tetanus-pertussis containing vaccine and measles containing vaccines) for each country and maybe health care worker influenza vaccine uptake for each country in Table 1.
Dear reviewer we did not find the requested information in the reference sources that we used. We do consider your proposal as extremely valuable and will try to identify comparative vaccination rate information in our future studies.
- Line 100: fines and penalties are rarely enforced: how do you know that what is your reference for this?
Added source 8. (p.2) Haverkate M, D'Ancona F, Giambi C, Johansen K, Lopalco PL, Cozza V, Appelgren E; VENICE project gatekeepers and contact points. Mandatory and recommended vaccination in the EU, Iceland and Norway: results of the VENICE 2010 survey on the ways of implementing national vaccination programmes. Euro Surveill. 2012 May 31;17(22):20183. doi: 10.2807/ese.17.22.20183-en. PMID: 22687916.
- 3.2. Policy measures at European level: I am uncertain how this section relates to the scoping review findings. Could the authors please outline how the findings such as “enhanced cooperation” came out of the scoping review? This section sounds to me a bit more suited to an editorial rather than actual findings of a scoping review. How did you come to these conclusions from the studies you looked at?
To better reflect the meaning the title of the subchapter was reformulated as: 3.2. Vaccination strengthening initiatives at European level
- Discussion:
- Lines 262-297: should these be part of the results rather than discussion?
Thank you, the text was removed to results section
- “Social justice” was not hinted at or mentioned in the results: it is first mentioned in discussion: what are the authors drawing in as part of their results to make the link to the discussion?
Yes, thank you. The social justice is discussed only from the point of view of the international collaboration. The whole paragraph was revised to be consistent with the results of the study.
The paragraph was revised as follows:
With knowledge, science, medicines, procedures, political will and international cooperation available, it is really an issue of justice and fairness to collectively tackle public health challenges. Upholding individual autonomy and freedom of choice in public health interventions in fact diminishes, if not entirely limits, individual freedom and choice on individual level because of the consequences of ill-health [37, 38].
- I really cannot understand the connection between the results and the discussion, and I think this whole section needs to be redone so that it matches the results. For example, the CPME survey findings: were they used as a part of the scoping review OR are they just used as part of the discussion and those findings not actually part of the results? If so, then why did the search not yield reference 26 (the CPME survey) as part of the scoping review sources?
Thank you. We moved the information for the CPME study at the results section and redesign the discussion.
Conclusion: What does this line mean: " There is no different approaches towards the vaccination policy for healthcare workers." Does it mean "this is not different than the approach for vaccination policy for health care workers?
Thank you – we revised the conclusion as follows:
The comparative analysis of vaccination policies across European countries underscores the diversity of approaches within the broader context of public health and individual choice. Each nation's policy reflects its unique cultural, legal, and healthcare considerations while striving to protect public health through vaccination. The nuanced strategies employed highlight the importance of tailored approaches that align with the specific needs and values of each society.
Comments on the Quality of English Language –
Please ensure you use the same tenses throughout and same passive or active voice throughout. For example Materials and methods has multiple tenses used, causing some confusion. Third person tenses are often incorrect and I wonder whether some past tenses were just missed in editing. There are also multiple grammatical errors throughout the document: please edit for grammar and tense use. Some simple examples that need correcting: you use both term immunisation as well as immunization: choose one. The whole manuscript needs a very detailed edit for appropriate language. - The language was revised by native English speaker
Reviewer 2 Report
Comments and Suggestions for Authors
Title: the authors describe their approach as a scoping review. Then I would suggest to include this information in the title.
Abstract and keywords: need some refinement after revision
Introduction: Line 33: it is not clear to me why the authors list plague as one of the diseases which have been eliminated in some regions of the world due to vaccination. To my knowldge, there is no public vaccination schedule against plague in Europe or elsewhere. Lines 33-36: herd immunity is clearly a very important aspect of vaccination policies but especially for tetanus and to a minor degree for hepatitis B and SARS-CoV-2 this concept is not the leading argument. Here self-protection is the main goal. Please consider to add this aspect. Lines 61-65: the study objectives are clearly stated but the results and discussion do not reflect these objectives clearly (see later comments).
Methods: Line 67: as the authors state that they used the methodology of a scoping review they should adhere to the proposed standards of such an approach (www.prisma-statement.org/scoping). Lines 71-78: the search strategy is not well described. Lines 79-80: the ECDC Vaccine Scheduler seems well designed, but as a clinician from Germany I noted that it contains factual mistakes on hepatitis B (it offers active and passive immunisation against hep B in neonates born by mothers with chronic hep B) and measles (will come back to this later). I am not in the position to check the calendars of the other countries but I would ask the authors to check directly with the national authorities if all information presented in the ECDC scheduler is correct.
Results: section 3.1.: table 1 does not provide much information. For the countries which employ mandatory policies the authors should give more precise information as to which vaccine is mandatory or not. This information should be given separately for children, adults and health professionals. Reading just the table it seems that several countries may have a mandatory policy for all vaccines .... . For Germany the only mandatory vaccine is measles. Lines 96-100 plus 140-143 plus 177-178: the case in Germany is much more complex than presented. For kindergarten attendance it is mandatory to have measles vaccination. But as kindergarten is not a compulsory institution for pre-school children, parents can opt out of kindergarten without any fines. On the contrary, school attendance in Germany is compulsory. For children with measles vaccination it is no problem to attend. For children without measles vaccination they must attend school, but parents are liable to be penalised with a fine. Healthworkers must provide proof of measles immunity, either by natural infection or 2 vaccine doses. Otherwise they cannot work in their profession. The same applies to adult staff in school and kindergarten. - Such a complex situation may exist in other EU countries, too. Please check again the rules for each country. They may differ from what is presented in lines 111-168. The whole section 3.1. needs refinement. At present it does not reflect the objectives of this paper as mentioned in lines 61-65, especially when looking at the regulations/policies for children and healthcare professionals. Section 3.2: it reads much better and provides a good overview of EU initiatives. Just one minor remark: the EU president is named Ursula von der Leyen.
Discussion: The discussion would need some re-organisation after revision of the previous sections. Lines 298-321: the discussion on bioethical aspects of vaccination policies seems somehow weak. It lacks some more references and should clearly list the pros and cons of voluntary versus mandatory vaccination policies for different vaccines and age groups. The authors may start with a search for Julian Savulescu and his ethics papers on vaccination policies. Lines 322-325: as reference 31 is missing it remains unclear what their statement on "anti-reductionist point of view" should describe. Lines 326-328: the discussion of the limitations and strengths of the study is very short, the remark about the omission of SARS-CoV-2 vaccines in the paper seems to be detached somehow.
References: reference 31 is missing in my version of the manuscript
I would suggest to have an additional look at these papers:
2: Haverkate M, D'Ancona F, Giambi C, Johansen K, Lopalco PL, Cozza V, Appelgren
E; VENICE project gatekeepers and contact points. Mandatory and recommended
vaccination in the EU, Iceland and Norway: results of the VENICE 2010 survey on
the ways of implementing national vaccination programmes. Euro Surveill. 2012
May 31;17(22):20183. doi: 10.2807/ese.17.22.20183-en. PMID: 22687916.
3: Paul KT, Loer K. Contemporary vaccination policy in the European Union:
tensions and dilemmas. J Public Health Policy. 2019 Jun;40(2):166-179. doi:
10.1057/s41271-019-00163-8. PMID: 30894672.
4: Bechini A, Boccalini S, Ninci A, Zanobini P, Sartor G, Bonaccorsi G, Grazzini
M, Bonanni P. Childhood vaccination coverage in Europe: impact of different
public health policies. Expert Rev Vaccines. 2019 Jul;18(7):693-701. doi:
10.1080/14760584.2019.1639502. Epub 2019 Jul 19. PMID: 31268739.
5: Siciliani L, Wild C, McKee M, Kringos D, Barry MM, Barros PP, De Maeseneer J,
Murauskiene L, Ricciardi W; members of the Expert Panel on Effective Ways of
Investing in Health. Strengthening vaccination programmes and health systems in
the European Union: A framework for action. Health Policy. 2020
May;124(5):511-518. doi: 10.1016/j.healthpol.2020.02.015. Epub 2020 Mar 6.
Erratum in: Health Policy. 2020 Sep;124(9):1041. doi:
10.1016/j.healthpol.2020.06.016. PMID: 32276852.
6: Valdecantos RL, Palladino R, Lo Vecchio A, Montella E, Triassi M, Nardone A.
Organisational and Structural Drivers of Childhood Immunisation in the European
Region: A Systematic Review. Vaccines (Basel). 2022 Aug 25;10(9):1390. doi:
10.3390/vaccines10091390. PMID: 36146467; PMCID: PMC9505321.
Please check the English grammar and writing, there a quite a few typos and grammatical errors.
Author Response
Comments and Suggestions for Autho
Dear Reviewer, thank for your valuable comments, below in red are explained the changes that we made through the text to answer to your queries.
Title: the authors describe their approach as a scoping review. Then I would suggest to include this information in the title.
We clarify at the methodology section our approach as follows below and decide not to change the title:
Policy review was undertaken with a focus on EU-level and national policies, using the webpages of the European Commission, the European Centre for Disease Prevention and Control (ECDC), and European-wide professional organizations of physicians. Regulatory and policy documents were identified and analyzed. The review was guided by four principal research questions: whether a unified approach to vaccination existed among EU member states, what vaccinations were mandatory or recommended, and whether there were specific provisions for healthcare professionals. As the primary focus were policy documents and EU-websites and relevant policy documents were identified, systematic literature review approach was not applied.
However, further search parameters were refined with the collaboration of a librarian from Emory Health Sciences. The following specific search terms were identified and utilized: “health care providers”, “communication”, “public health”, “informed consent”, and “bioethics”. The results of this search yielded a limited number of articles: "community engagement" returned 3 articles, "moral and political philosophy" yielded 9 articles, "informed consent in public health interventions" produced 7 articles, "public health ethics" resulted in 4 articles, and "informed consent in public health" yielded 6 articles.
Abstract and keywords: need some refinement after revision
Thank you - corrected
Introduction: Line 33: it is not clear to me why the authors list plague as one of the diseases which have been eliminated in some regions of the world due to vaccination. To my knowledge, there is no public vaccination schedule against plague in Europe or elsewhere. Lines 33-36: herd immunity is clearly a very important aspect of vaccination policies but especially for tetanus and to a minor degree for hepatitis B and SARS-CoV-2 this concept is not the leading argument. Here self-protection is the main goal. Please consider to add this aspect. Lines 61-65: the study objectives are clearly stated but the results and discussion do not reflect these objectives clearly (see later comments).
Thank you. Plaque was removed, self protection is added as follows:
Its worth mentioning that for some infectious diseases, especially for tetanus, and to a minor degree for hepatitis B and SARS-CoV-2 this concept is not the leading argument. Here self-protection is the main goal but still society have to be responsible to raise awareness.
Methods: Line 67: as the authors state that they used the methodology of a scoping review they should adhere to the proposed standards of such an approach (www.prisma-statement.org/scoping). Lines 71-78: the search strategy is not well described. Lines 79-80: the ECDC Vaccine Scheduler seems well designed, but as a clinician from Germany I noted that it contains factual mistakes on hepatitis B (it offers active and passive immunisation against hep B in neonates born by mothers with chronic hep B) and measles (will come back to this later). I am not in the position to check the calendars of the other countries but I would ask the authors to check directly with the national authorities if all information presented in the ECDC scheduler is correct.
Thank you – we explain better the methodology as policy review as stated under your previous question. We revised again the ECDC data and elaborated Table 1 by adding more data. We recognize the risk that some of the information at the ECDC might not reflect the current situation and add the date of search (accessed ECDC webpage in November), as well as pointed as limitation of the study. Further analyses should be done to validate the ECDC information with the national authorities.
Results: section 3.1.: table 1 does not provide much information. For the countries which employ mandatory policies the authors should give more precise information as to which vaccine is mandatory or not. This information should be given separately for children, adults and health professionals. Reading just the table it seems that several countries may have a mandatory policy for all vaccines .... . For Germany the only mandatory vaccine is measles. Lines 96-100 plus 140-143 plus 177-178: the case in Germany is much more complex than presented. For kindergarten attendance it is mandatory to have measles vaccination. But as kindergarten is not a compulsory institution for pre-school children, parents can opt out of kindergarten without any fines. On the contrary, school attendance in Germany is compulsory. For children with measles vaccination it is no problem to attend. For children without measles vaccination they must attend school, but parents are liable to be penalised with a fine. Health workers must provide proof of measles immunity, either by natural infection or 2 vaccine doses. Otherwise, they cannot work in their profession. The same applies to adult staff in school and kindergarten. - Such a complex situation may exist in other EU countries, too. Please check again the rules for each country. They may differ from what is presented in lines 111-168. The whole section 3.1. needs refinement. At present it does not reflect the objectives of this paper as mentioned in lines 61-65, especially when looking at the regulations/policies for children and healthcare professionals. Section 3.2: it reads much better and provides a good overview of EU initiatives. Just one minor remark: the EU president is named Ursula von der Leyen.
Thank you. We completely redesign the country’s comments. Once again we checked the information and changed the name of Mrs. Leyen. We do believe that at the time of search the ECDC database was correct, but probably does not contain all details for the countries policies. We also add this comment to the limitation.
Discussion: The discussion would need some re-organisation after revision of the previous sections. Lines 298-321: the discussion on bioethical aspects of vaccination policies seems somehow weak. It lacks some more references and should clearly list the pros and cons of voluntary versus mandatory vaccination policies for different vaccines and age groups. The authors may start with a search for Julian Savulescu and his ethics papers on vaccination policies. Lines 322-325: as reference 31 is missing it remains unclear what their statement on "anti-reductionist point of view" should describe. Lines 326-328: the discussion of the limitations and strengths of the study is very short, the remark about the omission of SARS-CoV-2 vaccines in the paper seems to be detached somehow.
Thank you. The discussion was reorganized and some new references were added as proposed by reviewer.
References: reference 31 is missing in my version of the manuscript – added reference 39
I would suggest to have an additional look at these papers:
2: Haverkate M, D'Ancona F, Giambi C, Johansen K, Lopalco PL, Cozza V, Appelgren
E; VENICE project gatekeepers and contact points. Mandatory and recommended
vaccination in the EU, Iceland and Norway: results of the VENICE 2010 survey on
the ways of implementing national vaccination programmes. Euro Surveill. 2012
May 31;17(22):20183. doi: 10.2807/ese.17.22.20183-en. PMID: 22687916.
Thank you – added as reference 7
3: Paul KT, Loer K. Contemporary vaccination policy in the European Union:
tensions and dilemmas. J Public Health Policy. 2019 Jun;40(2):166-179. doi:
10.1057/s41271-019-00163-8. PMID: 30894672.
Thank you – added as reference 13
4: Bechini A, Boccalini S, Ninci A, Zanobini P, Sartor G, Bonaccorsi G, Grazzini
M, Bonanni P. Childhood vaccination coverage in Europe: impact of different
public health policies. Expert Rev Vaccines. 2019 Jul;18(7):693-701. doi:
10.1080/14760584.2019.1639502.
Thank you – added reference 44
5: Siciliani L, Wild C, McKee M, Kringos D, Barry MM, Barros PP, De Maeseneer J,
Murauskiene L, Ricciardi W; members of the Expert Panel on Effective Ways of
Investing in Health. Strengthening vaccination programmes and health systems in
the European Union: A framework for action. Health Policy. 2020
May;124(5):511-518. doi: 10.1016/j.healthpol.2020.02.015. Epub 2020 Mar 6.
Erratum in: Health Policy. 2020 Sep;124(9):1041. doi:
10.1016/j.healthpol.2020.06.016. PMID: 32276852.
Thank you - Added reference 46
6: Valdecantos RL, Palladino R, Lo Vecchio A, Montella E, Triassi M, Nardone A.
Organisational and Structural Drivers of Childhood Immunisation in the European
Region: A Systematic Review. Vaccines (Basel). 2022 Aug 25;10(9):1390. doi:
10.3390/vaccines10091390. PMID: 36146467; PMCID: PMC9505321.
Thank you – added as reference 45
Comments on the Quality of English Language
Please check the English grammar and writing, there a quite a few typos and grammatical errors. – Thank you, the text was revised by native speaker.
Reviewer 3 Report
Comments and Suggestions for Authors
A review of the manuscript entitled “Unity in diversity and diversity in unity- vaccination policies in EU countries”
1. Authors are suggested to proofread the manuscript after addressing all comments to avoid any typo, grammatical, and lingual mistakes and errors. For example, “… as defined by WHO “refers to …” on page 2 line 59; “… underscoresthe country's respect…” on page 5 line 167; “Althougth” on page 5 line 178.
2. (page 2, lines 58-60): This sentence “Vaccine hesitancy as defined by WHO refers to delay in acceptance or refusal of vaccines despite availability of vaccination services.” will be more significant if it’s also supported by another relevant study. Please check https://doi.org/10.52225/narra.v1i3.57
3. (page 8, line 298): What is the meaning of “art. 14” Please give an explanation about it.
4. (page 1, lines 27-29): This sentence also shares the same idea with a study entitled “Global prevalence and determinants associated with the acceptance of monkeypox vaccination” Kindly include this reference to make the sentence stronger.
5. Overall, this manuscript has shown a sophisticated review of vaccination policies in the EU, using lots of relevant references in related areas. The use of good English has also been applied throughout the manuscript. However, I have some minor comments, as already mentioned above, to increase the clarity of this review manuscript. Thank you.
Author Response
Dear Reviewer, thank for your valuable comments, below are explained the changes that we made through the text to answer to your queries.
A review of the manuscript entitled “Unity in diversity and diversity in unity- vaccination policies in EU countries”
- Authors are suggested to proofread the manuscript after addressing all comments to avoid any typo, grammatical, and lingual mistakes and errors. For example, “… as defined by WHO “refers to …” on page 2 line 59; “… underscores the country's respect…” on page 5 line 167; “Althougth” on page 5 line 178.
Thank you – we did the necessary revisions and made the necessary changes with the support of the native speaker.
- (page 2, lines 58-60): This sentence “Vaccine hesitancy as defined by WHO refers to delay in acceptance or refusal of vaccines despite availability of vaccination services.” will be more significant if it’s also supported by another relevant study. Please check https://doi.org/10.52225/narra.v1i3.57
Thank you – added as reference 18
- (page 8, line 298): What is the meaning of “art. 14” Please give an explanation about it.
The sentence was changed as follows: From a bioethical perspective, the issue on social responsibility and health is discussed in art. 14 from the Universal Declaration of Bioethics and Human
- (page 1, lines 27-29): This sentence also shares the same idea with a study entitled “Global prevalence and determinants associated with the acceptance of monkeypox vaccination” Kindly include this reference to make the sentence stronger.
Thank you – added as reference 7
- Overall, this manuscript has shown a sophisticated review of vaccination policies in the EU, using lots of relevant references in related areas. The use of good English has also been applied throughout the manuscript. However, I have some minor comments, as already mentioned above, to increase the clarity of this review manuscript. Thank you.
Thank you for your proposals and new references – they were added to the text and clarify the issues in the article.
Round 2
Reviewer 2 Report
Comments and Suggestions for Authors
Thanks to the authors for all their efforts. They tried hard to respond to the suggested changes. The manuscript reads now much better than the first version. Nevertheless, I am not convinced that the details in table 1 and the conclusions drawn from it are correct for all countries. For Germany, it is still not correct to state that health professionals have to undergo mandatory vaccinations. This applies only to measles vaccination. Anyways, the ethical discussion has been improved considerably by including papers from some of the leading experts in this field.
Some minor remarks:
References: ref 9 and 21 are identical, ref 19 and 22 are identical
There are still several typos and grammar errors.
Author Response
Reviewer 1
Thanks to the authors for all their efforts. They tried hard to respond to the suggested changes. The manuscript reads now much better than the first version. Nevertheless, I am not convinced that the details in table 1 and the conclusions drawn from it are correct for all countries. For Germany, it is still not correct to state that health professionals have to undergo mandatory vaccinations. This applies only to measles vaccination. Anyways, the ethical discussion has been improved considerably by including papers from some of the leading experts in this field.
Thank you for the guidance for improving the manuscript. Information in Table 1 has been obtained through ECDC, CPME and literature review. We have added “measles” in the Table for Germany’s entry. We have further contacted the German Medical Association, however, reply has not been received within the 5-day time limit. Please accept apologies.
Some minor remarks:
References: ref 9 and 21 are identical, ref 19 and 22 are identical
Thank you, the references have been updated and corrected
Corrected
There are still several typos and grammar errors.
Hope to have addressed those too with the help of a native English speaker.
Reviewer 3 Report
Comments and Suggestions for Authors
First, I would like to commend the authors for addressing my concerns. This manuscript has significantly improved compared to its previous version. However, the revised version with tracked changes is somehow hampering me from an enjoyable reading process. Perhaps in the future, authors could upload two versions of the revised manuscript, one with tracked changes and one without, to facilitate the reviewer's reading process. Thanks!
Author Response
Reviewer 2
Comments and Suggestions for Authors
First, I would like to commend the authors for addressing my concerns. This manuscript has significantly improved compared to its previous version. However, the revised version with tracked changes is somehow hampering me from an enjoyable reading process. Perhaps in the future, authors could upload two versions of the revised manuscript, one with tracked changes and one without, to facilitate the reviewer's reading process. Thanks!
Dear Reviewer,
We are sorry for the misunderstanding, but we really uploaded 2 versions of the manuscript. The one was a clear version with main text and the second was a supplementary file in track changes. We really don't know why the system shows only the track version file. Once again please accept our sincere apologies.